# Cross-Reactive Fc-Mediated Antibody Responses to Influenza HA Stem Region in Human Sera Following Seasonal Vaccination

**DOI:** 10.3390/vaccines13020140

**Published:** 2025-01-28

**Authors:** Ayae Nishiyama, Takuto Nogimori, Yuji Masuta, Tomoka Matsuura, Tetsuo Kase, Kyoko Kondo, Satoko Ohfuji, Yu Nakagama, Natsuko Kaku, Sachie Nakagama, Yuko Nitahara, Yoshimasa Takahashi, Hiroshi Kakeya, Yasutoshi Kido, Wakaba Fukushima, Takuya Yamamoto

**Affiliations:** 1Laboratory of Precision Immunology, Center for Intractable Diseases and ImmunoGenomics, National Institutes of Biomedical Innovation, Health, and Nutrition, Osaka 567-0085, Japan; anishiyama@nibiohn.go.jp (A.N.); tnogimori@nibiohn.go.jp (T.N.); y-masuta@nibiohn.go.jp (Y.M.); 2Department of Public Health, Graduate School of Medicine, Osaka Metropolitan University, Osaka 545-8585, Japan; h21594d@omu.ac.jp (T.M.); kasetetsuo@omu.ac.jp (T.K.); satoko@omu.ac.jp (S.O.); wakaba@omu.ac.jp (W.F.); 3Research Center for Infectious Disease Sciences, Graduate School of Medicine, Osaka Metropolitan University, Osaka 545-8585, Japan; nakagama.yu@omu.ac.jp (Y.N.); kaku@omu.ac.jp (N.K.); d21385x@omu.ac.jp (S.N.); e21499e@omu.ac.jp (Y.N.); kakeya@omu.ac.jp (H.K.); kidoyasu@omu.ac.jp (Y.K.); 4Management Bureau, Osaka Metropolitan University Hospital, Osaka 545-8585, Japan; kyou@omu.ac.jp; 5Department of Virology and Parasitology, Graduate School of Medicine, Osaka Metropolitan University, Osaka 545-8585, Japan; 6Research Center for Drug and Vaccine Development, National Institute of Infectious Diseases, Tokyo 162-8640, Japan; ytakahas@niid.go.jp; 7Department of Infection Control Science, Graduate School of Medicine, Osaka Metropolitan University, Osaka 545-8585, Japan; 8Osaka International Research Center for Infectious Diseases, Osaka Metropolitan University, Osaka 545-0051, Japan; 9Laboratory of Aging and Immune Regulation, Graduate School of Pharmaceutical Sciences, Osaka University, Osaka 565-0871, Japan; 10Department of Virology and Immunology, Graduate School of Medicine, Osaka University, Osaka 565-0871, Japan

**Keywords:** influenza virus, vaccine, antibody-dependent cellular cytotoxicity, antibody-dependent cellular phagocytosis

## Abstract

**Background:** Current influenza A vaccines primarily induce neutralizing antibodies targeting the variable hemagglutinin (HA) head domain, limiting their effectiveness against diverse or emerging influenza A virus (IAV) subtypes. The conserved HA stem domain, particularly the long α-helix (LAH) epitope, is a focus of universal vaccine research due to its cross-protective potential. Additionally, Fc-mediated functions such as antibody-dependent cellular cytotoxicity (ADCC) and antibody-dependent cellular phagocytosis (ADCP) are recognized as important protective immune mechanisms. This study evaluated IgG responses to the HA head, stem, and LAH regions and assessed cross-reactive potential through neutralization, ADCC, and ADCP assays. **Methods:** IgG responses to the HA head, stem, and LAH regions were measured in vaccinated individuals. Functional assays were conducted for neutralization, ADCC, and ADCP to evaluate the association between antibody levels and immune function. **Results:** The results showed that HA head-specific IgG increased significantly after vaccination in 50 individuals, whereas stem-specific IgG increased by 72% and LAH-specific IgG by 12–14%. Among the induced antibody subclasses, IgG1 was predominantly increased. Neutralization titers were detected in viruses of the same strain as the vaccine strain, but not in classical or pandemic strains (H5N1, H7N9). HA stem-specific IgG1 antibody titers showed a significant correlation with ADCC/ADCP activity breadth, but no correlation was observed with neutralization breadth. **Conclusions:** These findings suggest that although current influenza vaccines can induce HA stem-targeted cross-reactive antibodies, their quantity may be insufficient for broad cross-protection, underscoring the need for improved vaccine strategies.

## 1. Introduction

It has been reported that the protective immune responses against Influenza A viruses (IAVs) elicited by current influenza vaccines depend on vaccine-induced neutralizing antibodies that primarily target hemagglutinin (HA) [1,2,3]. The HA proteins of IAVs are the main targets of neutralizing antibodies and undergo continuous evolution owing to selective pressure from antibody responses, which are directed primarily at the distal membrane receptor-binding subdomain of the molecule [4]. Genetically, HA proteins of IAVs are classified into 18 subtypes and grouped into two groups [5]. Antibodies induced by vaccination or infection primarily neutralize homologous strains within the same subtype. IAV HA proteins consist of two major domains: the head and stem. The HA head domain is the most mutable region in IAV proteins, which limits the cross-reactivity of vaccine-induced antibodies to other IAV subtypes or antigenic variants. Consequently, the current influenza vaccines are effective only against a narrow range of IAV strains with HA epitopes that are antigenically similar to the vaccine strain. This limitation has reduced their effectiveness against other IAV subtypes, including in future pandemics. HA proteins of IAVs contain multiple classes of antibody epitopes, some of which are relatively conserved among strains [1,6,7,8,9,10,11]. In contrast, conformational epitopes in the HA stem domain are conserved across different subtypes and are the primary targets of several universal influenza vaccines under clinical trials [1,6,7,12]. One notable epitope in the long α-helix region of the HA stem domain is known as the long α-helix (LAH) epitope, which is linear and occluded in the HA trimer structure, distinguishing it from the conformational stem epitope exposed in the native trimer structure. Antibody responses against the LAH epitope are known to increase at local sites following viral infection as the LAH epitope becomes exposed in the HA form after membrane fusion [13,14,15].

Recently, IgG Fc-mediated functions of antibody-dependent cellular cytotoxicity (ADCC) and antibody-dependent cellular phagocytosis (ADCP) have garnered attention as protective immune responses against IAV infection [16,17,18]. The effectiveness of ADCC and ADCP depends on the affinity of IgG for both the antigen epitope and the activated Fcγ receptor (FcγR). In humans, IgG1 has the highest affinity for FcγR binding, followed by IgG3, IgG2, and IgG4, in that order [19]. Therefore, the balance between the IgG subclasses is an important factor in the effectiveness of universal influenza vaccines. Indeed, multiple lines of evidence have correlated IgG subclass profiles with the cross-protective abilities of influenza vaccines [7,20,21,22,23]. Antibodies targeting the head–interface epitope bind to multiple HA subtypes across both group 1 (e.g., H1 and H5) and group 2 (e.g., H3 and H7) and engage in Fc-mediated effector functions through non-neutralizing IgG to confer multivariant protection. Similarly, cross-protective IgG antibodies against the LAH epitope are non-neutralizing and, similar to antibodies against the HA stem domain, provide protection via IgG Fc-mediated effector function [10,11,13].

Therefore, this study aimed to comprehensively evaluate the capacity of antibody-mediated cross-reactive protective responses following vaccination with the current seasonal influenza vaccines. First, IgG antibody titers against the HA head, stem, and LAH regions of the vaccine strains induced by vaccination were measured. Next, the antibody function was assessed using neutralization titers and ADCC/ADCP activity as parameters. The results showed that the IgG antibodies against the HA head and stem regions of the vaccine strains were significantly elevated after vaccination. In contrast, antibodies against the LAH region, a highly conserved epitope, showed no correlation with the neutralization breath scores. However, the ADCC and ADCP breath scores were significantly correlated with IgG1 levels against the HA stem region, suggesting that the HA stem-targeted antibodies induced by the current vaccine may exhibit cross-reactivity. Despite these results, it is generally accepted that the antibodies elicited by current seasonal influenza vaccines are not expected to provide cross-protective effects [24]. Therefore, although cross-reactive antibodies may be induced, their quantities may be insufficient to achieve a protective effect.

## 2. Materials and Methods

### 2.1. Study Design and Subjects

A total of 50 individuals were enrolled from healthcare workers at Osaka Metropolitan University Hospital, staff of the Osaka City Health Bureau, and faculty members and students from the School of Medicine and the Graduate School of Nursing at Osaka Metropolitan University in Japan. Blood samples were obtained one month prior to and one month following vaccination. Samples were processed using the INSEPACK II-D (SEKISUI MEDICAL Co., Ltd., Tokyo, Japan) kit to separate the serum, which was subsequently stored at −80 °C. Details of the donors included in this study are provided in Appendix A. The vaccine used in this study was administered to participants following their explicit consent. The vaccine manufacturers included Denka Co., Ltd. (Tokyo, Japan); the Research Foundation for Microbial Diseases of Osaka University; KM Biologics Co., Ltd. (Kumamoto, Japan); and DAIICHI SANKYO Co., Ltd. (Tokyo, Japan) All vaccines were egg-based, inactivated, split influenza vaccines using standardized domestic strains (A; A/Victoria/1/2020 [IVR-217] and A/Darwin/9/2021 [SAN-010] and B; B/Phuket /3073/2013 and B/Austria/1359417/2021 [BVR-26]) approved by the national regulatory authority. There were no differences in formulation among the vaccines, and the method of administration was confirmed to be consistent across all participants.

This study was approved by the Institutional Review Boards of the National Institutes of Biomedical Innovation, Health, and Nutrition (permit number: 292m; 25 August 2023), Ethical Committee of Osaka Metropolitan University Graduate School of Medicine (permit number: 2023-0032K; 2 October 2023) and Osaka Metropolitan University Hospital Certified Review Board (permit numbers: OCU010E and OCU013E; 1 March 2021 and 11 January 2022). All the experiments were performed in accordance with the principles of the Declaration of Helsinki. All volunteers provided written informed consent prior to enrollment.

### 2.2. Virus Strains

The virus strains were used for neutralization assays, ADCC assays, and ADCP assays: H1N1 (A/Puerto Rico/8/1934, A/California/04/2009, A/Guangdong-Maonan SWL1536/2019, A/Victoria/1/2020 (IVR-217)), H5N1 (A/Vietnam/1194/2004 (NIBRG-14)), H3N2 (A/X-31, A/Guizhou/54/1989, A/Hong Kong/2671/2019, A/Darwin/9/2021 (SAN-010)), and H7N9 (A/Anhui/1/2013 (NIBRG-268)). A/Victoria/1/2020 (IVR-217) and A/Darwin/9/2021 (SAN-010) are identical to the strains included in the administered vaccine. Each virus was propagated in the Madin–Darby canine kidney (MDCK) cell line, and supernatants were collected.

### 2.3. Enzyme-Linked Immunosorbent Assay (ELISA)

The levels of serum IgG antibodies targeting recombinant HA (rHA) proteins derived from vaccine strains, as well as H1 LAH and H3 LAH peptides (Appendix A), were assessed using an ELISA method as previously reported [14]. The rHA proteins used in the ELISA assays were produced in-house. Specifically, synthetic cDNA for each antigen was obtained through artificial gene synthesis and inserted into a plasmid under the control of a CMV promoter. This plasmid was subsequently transfected into Expi293 cells, which were cultured for five days post-transfection. After the culture period, the supernatants were collected, and the target antigen proteins were purified using His tag affinity chromatography. Peptides rather than full-length proteins were used to specifically evaluate the LAH region, as the latter would also include non-specific regions, potentially confounding the results. In brief, 96-well plates were prepared by coating them with 50 µL of rHA protein solution (0.5 μg/mL) and LAH peptide solution (0.5 μg/mL). Afterward, the plates were washed four times with 0.05% Tween20 in sterile phosphate-buffered saline (PBS-T), blocked using ChonBlock (Chondrex, Woodinville, WA, USA) at room temperature for 2 h, and incubated with diluted serum samples for another 2 h at room temperature. Next, the plates were washed and incubated for 1 h with various secondary anti-human IgG antibodies, including mouse anti-Human IgG Fc-BIOT (SouthernBiotech, USA), mouse anti-Human IgG1 Fc secondary antibody-HRP (Thermo Fisher Scientific, Waltham, MA, USA), mouse Anti-Human IgG2 Fc-BIOT (SouthernBiotech, Birmingham, AL, USA), mouse Anti-Human IgG3 Hinge-BIOT (SouthernBiotech, Birmingham, AL, USA), or mouse Anti-Human IgG4 Fc-BIOT (SouthernBiotech, Birmingham, AL, USA). For total IgG, IgG2, IgG3, and IgG4 detection, the plates were further washed and incubated with HRP-labeled streptavidin (Thermo Fisher Scientific, Waltham, MA, USA) for 1 h at room temperature. Following another round of washing with PBS-T, the plates were treated with the TMB Microwell Peroxidase Substrate System reagent (KPL, Gaithersburg, MD, USA) to initiate the colorimetric reaction, following the manufacturer’s instructions. The reaction was terminated by adding 2 N H_2_SO_4_, and the absorbance at 450 nm (OD450) was measured. Serum IgG levels were quantified using reference IgG clones targeting the conserved rHAs, stem regions, and LAH epitopes (FI6 clone [7] for rHAs and CR9114 clone [1] for stem, H1, and H3 LAH).

### 2.4. In Vitro Neutralization Assay

Neutralizing antibody titers were assessed using a previously established method with the MDCK cell line [25,26]. Prior to the neutralization assays, serum samples were treated with receptor-destroying enzyme II (Denka Seiken, Tokyo, Japan) following the manufacturer’s instructions to remove non-specific serum inhibitors. Serially diluted serum was pre-incubated with either H1N1, H3N2, H5N1, or H7N9 viruses (100 TCID50) and subsequently added to MDCK cells in the presence of 5 μg/mL acetyl trypsin (Sigma, St. Louis, MO, USA). Each plate included eight control wells containing either virus alone or diluent. The plates were incubated for 3 days at 37 °C in a humidified atmosphere with 5% CO_2_. After incubation, all wells were fixed with 10% formalin in phosphate-buffered saline (PBS) for 30 min at room temperature and stained with naphthol blue black. Neutralization titers were defined as the reciprocal of the highest serum dilution that prevented cytopathic effects.

### 2.5. ADCC Assay and ADCP Assay

ADCC and ADCP assays were conducted using Jurkat-Lucia™ NFAT-CD16 Cells for ADCC and Jurkat-Lucia™ NFAT-CD32 Cells for ADCP (both from Invivogen, San Diego, CA, USA), following the manufacturer’s protocols. In summary, MDCK cells were plated in flat-bottom 96-well cell culture plates (Corning, Steuben County, NY, USA) at a density of 3.0 × 10^4^ cells per well and incubated overnight at 37 °C with 5% CO_2_. The next day, the cells were infected with each influenza virus at a concentration of 2 × 10^4^ PFU/mL and incubated again at 37 °C with 5% CO_2_. After 12–16 h, the medium was replaced with serially diluted serum prepared in assay buffer, and effector cells were added. Following an 8 h incubation, Quanti-Luc™ reagent and substrate (Invivogen, San Diego, CA, USA) were added, and luminescence was measured using an EnSpire Multimode Plate Reader (PerkinElmer, Waltham, MA, USA). ADCC and ADCP activities were defined as the 50% Fc-mediated effect of serum dilution achieved using Prism 8.4.3 software (GraphPad Software Inc., Boston, MA, USA).

### 2.6. Statistical Analysis

Statistical analyses were performed using Prism 8.4.3 software (GraphPad Software Inc., Boston, MA, USA). Non-parametric statistical methods, including the Wilcoxon matched-pair signed-rank tests, were applied as they are appropriate for datasets with unknown or non-normal distributions, ensuring robust and reliable analysis. Correlations were calculated using the nonparametric Spearman’s rank test. The significance of the differences between groups is indicated by bars and symbols as follows: * *p* < 0.05, ** *p* < 0.01, *** *p* < 0.001, and **** *p* < 0.0001.

## 3. Results

### 3.1. Characteristics of Serum Antibodies to Each HA Region Induced by Seasonal Influenza Vaccination

The function of antibodies against the HA protein of influenza A viruses depends on the specific region they recognize [27]. Therefore, we investigated the regions of the HA protein that are targeted by antibodies induced by the current seasonal influenza vaccine and characterized the antibodies that recognize these regions.

Initially, total IgG antibody titers against the HA head, stem, and LAH regions of the vaccine strains were measured using serum samples collected at two time points (pre-vaccination and 1 month post-vaccination) from 50 individuals vaccinated with the November 2022 seasonal influenza vaccine. The results showed that the total IgG titers specific to the HA head regions of both the H1 and H3 subtypes were significantly higher post-vaccination than pre-vaccination in all 50 individuals (*p* < 0.0001 for both) (Figure 1A).

Next, anti-H1 HA stem-specific total IgG was measured using a trimeric protein probe derived from the H1 NC strain [28] and was found to be significantly elevated post-vaccination in 36 of 50 individuals (72%) (*p* = 0.0002) (Figure 1A). Anti-H1 LAH-specific and anti-H3 LAH-specific total IgG levels were measured using peptides specific to the H1 and H3 LAH epitopes, respectively. In contrast to the anti-HA stem antibody, anti-H1 LAH-specific total IgG increased in only 6 of 50 individuals (12%), and anti-H3 LAH-specific total IgG increased in only 7 of 50 individuals (14%) post-vaccination (Figure 1A).

Furthermore, the affinity between IgG and FcγR plays an important role in protection against IAV infection in vivo and depends mainly on the Fc region. Among human IgG subtypes, IgG1 and IgG3 have high affinities for the FcγR [29,30]. Therefore, we analyzed the subclasses of each antigen-specific antibody to determine the IgG subclasses that were induced by the current seasonal influenza vaccine. Anti-H1 HA head-specific IgG1 was significantly higher post-vaccination-than pre-vaccination in all 50 individuals (*p* < 0.0001) (Figure 1B). In contrast, anti-H1 HA head-specific IgG2 levels were higher post-vaccination than pre-vaccination in 13 of the 50 individuals (26%) (Figure 1B). Anti-H3 HA head-specific IgG1 levels were significantly higher post-vaccination than pre-vaccination in 32 of 50 (64%) individuals (*p* < 0.0001) (Figure 1C). In contrast, anti-H3 HA head-specific IgG2 was higher post-vaccination than pre-vaccination in only 2 of the 50 individuals (4%) (Figure 1C). Anti-H1 HA stem-specific IgG1 levels were significantly elevated post-vaccination in 40 of 50 individuals (80%) (*p* = 0.0001) (Figure 1D). Anti-H1 HA stem-specific IgG2 was below the detection limit in 41 of 50 individuals. (Figure 1D). The levels of anti-H1 HA head-specific IgG3, anti-H3 HA head-specific IgG3, anti-H1 HA stem-specific IgG3, anti-H1 HA head-specific IgG4, anti-H3 HA head-specific IgG4, and anti-H1 HA stem-specific IgG4 were all below the detection limit.

### 3.2. Cross-Reactive Neutralizing Antibody Activity Induced by Current Seasonal Influenza Vaccination in Serum

Subsequently, to determine whether the current vaccine induced cross-reactive neutralizing antibodies, we measured neutralizing antibody titers against four H1N1, four H3N2, one H5N1, and one H7N9 virus strains from groups 1 and 2, including the vaccine strains. Neutralizing antibody titers were assessed for each of the 10 viral strains. To quantify the serum neutralizing titers, scores were assigned based on the neutralizing antibody titer of each virus strain (score 1 for titers between 21 and 80, score 2 for titers between 81 and 640, score 3 for titers between 641 and 1280, and score 4 for titers > 1280) (Appendix A). The neutralizing titers against viruses of the same strain as the H1 and H3 vaccine strains were divided into two categories: score 1 for 56% and 62%, respectively, and score 2 for 44% and 38%, respectively, for all 50 individuals. For A/Guangdong-Maonan SWL1536/2019 (H1N1), which is similar in age to the vaccine strain, 26% of the individuals had a score of 1, 16% had a score of 2, and 2% had a score of 3. For A/Hong Kong/2671/2019 (H3N2), 32% of individuals had a score of 1, 54% had a score of 2, 4% had a score of 3, and 4% had a score of 4, indicating higher scores for these strains than for the vaccine strain. For A/California/04/2009 (H1N1), which is more than 10 years older than the vaccine strain, 90% of individuals had a score of 1, and 2% of individuals had a score of 2. For A/Guizhou/54/1989 (H3N2), the results indicated a relatively high susceptibility to neutralization: 26% of the individuals scored 1, 62% scored 2, 6% scored 3, and 2% scored 4. In contrast, neutralizing antibody titers against the classic strains A/Puerto Rico/8/1934 (H1N1) and A/X-31 (H3N2) were below the detection limits in all 50 individuals. Furthermore, no neutralizing antibody titers were detected against pandemic influenza strains A/Vietnam/1194/2004 (H5N1) or A/Anhui/1/2013 (H7N9).

To quantify cross-reactive neutralization antibody activity based on these data, H1N1 and H5N1 viruses were classified as group 1, and H3N2 and H7N9 as group 2. The cross-reactive neutralization antibody activity for each group was calculated as the neutralization group 1 breath score, the neutralization group 2 breath score, and the combined score for group 1 and group 2. This combined score, referred to as the neutralization group 1 + 2 breadth score, was then calculated (Appendix A). The correlation between cross-reactive neutralization antibody activity and serum levels of anti-H1 HA-specific, anti-H3 HA-specific, or anti-H1 HA stem-specific IgG was examined. The results showed no significant correlation between the neutralization group 1 breath score and the anti-H1 HA head-specific, anti-H3 HA head-specific, or anti-H1 HA stem-specific total IgG titers against the vaccine strain (*p* = 0.2367, *p* = 0.1198, and *p* = 0.0677, respectively) (Figure 2A). Similarly, no correlation was found between neutralization the group 2 breath score and anti-H1 HA head-specific, anti-H3 HA head-specific, or anti-H1 HA stem-specific total IgG titers against the vaccine strains (*p* = 0.3739, *p* = 0.3691, and *p* = 0.9923, respectively) (Figure 2B).

These results indicated that although neutralizing antibodies against viral strains of similar age to the vaccine strain were detected to some extent in sera after vaccination with the current influenza vaccine, neutralizing antibodies against classical strains, H5N1 and H7N9, were not detected, indicating that the avidity of cross-reactive neutralizing antibodies was limited.

### 3.3. Cross-Reactive ADCC Activity Induced by Current Seasonal Influenza Vaccination in Serum

To investigate the cross-protective effects of HA-specific antibodies mediated by the Fc region, which is considered an important function beyond the neutralizing activity of cross-reactive antibodies, ADCC activity was measured using 10 viral strains, similar to those used for the measurement of neutralizing antibody titers. To quantify ADCC activity, the measured values for each viral strain were calculated as the Fold of Induction: *Fold of Induction* = (*RLU_induced_* − *RLU_background_*)/(*RLU_no antibody control_* − *RLU_background_*). The relationship between fold induction and the antibody dilution factor was analyzed using the following formula: The curve was fitted using curve-fitting software (GraphPad Software Inc., Boston, MA, USA), and the EC_50_ of the antibody response was determined (Appendix A). ADCC activity against the virus strain matching the H1 vaccine strain was observed in 44 of 50 individuals: 38% of individuals scored 1, 42% scored 2, 6% scored 3, and 2% scored 4. ADCC activity against the H3 vaccine strain was detected in only 12 of 50 individuals, with 8% scoring 1, 4% scoring 2, and 12% scoring 4. This indicates that ADCC activity was higher in the H1 strain than in the H3 strain. Next, ADCC activity against A/Guangdong-Maonan SWL1536/2019 (H1N1), a viral strain similar in age to the vaccine strain, was detected in 49 of the 50 individuals. The distribution of the scores was as follows: 16% scored 1, 56% scored 2, 22% scored 3, and 6% scored 4. For A/Hong Kong/2671/2019 (H3N2), ADCC activity was detected in all 50 individuals; 4% of the individuals scored 1, 46% scored 2, 24% scored 3, and 26% scored 4, indicating high ADCC activity.

In contrast, for A/California/04/2009 (H1N1), ADCC activity was observed in only 8 out of 50 individuals, all of whom scored 1. For A/Guizhou/54/1989 (H3N2), ADCC activity was detected in all 50 individuals, with 18% scoring 1, 68% scoring 2, and 14% scoring 3. Similarly, the classic strain A/Puerto Rico/8/1934 (H1N1) showed ADCC activity in 4 of the 50 individuals, but with low scores: 6% scored 1 and 2% scored 2. In contrast, ADCC activity against A/X-31, a classic H3N2 strain, was observed in 49 of 50 individuals, with 14% scoring 1, 58% scoring 2, 24% scoring 3, and 2% scoring 4. For A/Vietnam/1194/2004 (H5N1), ADCC activity was observed in 3 out of 50 individuals, with 28% scoring 1, 50% scoring 2, 14% scoring 3, and 2% scoring 4. These results indicated no relationship between the age of the vaccine strain and the age of the virus for ADCC activity.

To quantify cross-reactive ADCC activity based on these data, H1N1 and H5N1 viruses were classified as group 1 and H3N2 and H7N9 as group 2 in the same classification used for the measurement of neutralizing activity. The cross-reactive ADCC activity for each group was calculated as the ADCC group 1 breath score, ADCC group 2 breath score, and combined ADCC group 1 + 2 breath score, which was the sum of group 1 and group 2 points (Appendix A).

Next, we examined the correlation between cross-reactive ADCC activity and serum levels of anti-H1 HA-specific, anti-H3 HA-specific, and anti-H1 HA stem-specific IgG. A significant correlation was observed between the ADCC group 1 breath score and anti-H1 HA head-specific total IgG and IgG1 against the vaccine strain (r = 0.3394, *p* = 0.0159 and r = 0.3509, *p* = 0.0125, respectively). However, no correlation was found for IgG2 (r = 0.02798, *p* = 0.8471) (Figure 3A). In contrast, no correlation was observed between the ADCC group 2 breadth score and anti-H3 HA head-specific total IgG or IgG1 levels against the vaccine strain (r = 0.1161, *p* = 0.4220 and r = 0.1615, *p* = 0.2626, respectively) (Figure 3B). The ADCC group 1 breath score showed a significant correlation with the total anti-H1 HA stem-specific IgG and IgG1 titers (r = 0.5912, *p* < 0.0001, and r = 0.5592, *p* = 0.0001, respectively) (Figure 3C).

These results indicated that the current influenza vaccine may induce cross-reactive IgG antibodies with Fc-mediated ADCC functions against a wide range of viral strains in some individuals.

### 3.4. Cross-Reactive ADCP Activity Induced by Current Seasonal Influenza Vaccination in Serum

Next, we performed an ADCP assay using 10 viral strains to calculate ADCP and ADCC activities. First, to quantify ADCP activity, we calculated the measured values obtained for various viruses as *Fold of Induction* = (*RLU_induced_* − *RLU_background_*)/(*RLU_no antibody control_* − *RLU_background_*). The relationship between fold induction and the antibody dilution factor was then analyzed. Curves were fitted using curve-fitting software (GraphPad Prism, USA) to determine the EC_50_ of the antibody responses. Scores were assigned based on the ADCP activity for each viral strain as follows: 1 for titers between 11 and 100, 2 for titers between 101 and 500, 3 for titers between 501 and 1000, and 4 for titers > 1000 (Appendix A). ADCP activity against the viral strain matching the H1 vaccine strain was observed in 2 of 50 individuals, with 4% scoring 1. Against the viral strain matching the H3 vaccine strain, ADCP activity was observed in 18 of the 50 individuals, with 30% scoring 1 and 6% scoring 2. ADCP activity against A/Guangdong-Maonan SWL1536/2019 (H1N1), a viral strain of similar age as the vaccine strain, was detected in 41 of 50 individuals, with 50% scoring 1, 28% scoring 2, 2% scoring 3, and 2% scoring 4. For A/Hong Kong/2671/2019 (H3N2), ADCP activity was observed in 36 of 50 individuals, with 44% scoring 1, 26% scoring 2, and 2% scoring 3, showing a trend toward higher activity than that of the vaccine strain, similar to ADCC activity. ADCP activity against A/California/04/2009 (H1N1) was observed in 10 of 50 individuals, with 4% scoring 1, 8% scoring 2, and 8% scoring 4. For A/Guizhou/54/1989 (H3N2), ADCP activity was observed in 42 out of 50 individuals, with 64% scoring 1, 18% scoring 2, and 2% scoring 3. For the classic strain A/Puerto Rico/8/1934 (H1N1), ADCP activity was observed in 5 of 50 individuals, with 6% scoring 1 and 4% scoring 2, similar to ADCC activity. For the classic H3N2 strain A/X-31, ADCC activity was observed in 25 of 50 individuals, with 46% scoring 1, 2% scoring 2, and 2% scoring 4. ADCP activity was observed in 25 of the 50 individuals with A/X-31, the classic strain of H3N2, with 46% scoring 1, 2% scoring 2, and 2% scoring 4. For A/Vietnam/1194/2004 (H5N1), ADCP activity was observed in 36 of 50 individuals, with 34% scoring 1 and 32% scoring 2. A/Anhui/1/2013 (H7N9) was active in 13 of 50 individuals, but all showed low activity, scoring only 1 (Appendix A). These results indicate that the ADCP activity of cross-reactive IgG antibodies induced by vaccination differed between viral strains, regardless of the year of virus isolation. To quantify cross-reactive ADCP activity based on these data, H1N1 and H5N1 viruses were classified as group 1 and H3N2 and H7N9 as group 2, following the same classification used for neutralizing activity measurements. Cross-reactive ADCP activity scores were calculated as the ADCP group 1 breadth score, ADCP group 2 breadth score, and ADCP group 1 + 2 breadth score, representing the sum of group 1 and group 2 points (Appendix A). To verify the cross-reactivity of the ADCP activity of antibodies induced by vaccination, we first divided the viruses into groups 1 and 2 and calculated the ADCP activity scores for each group (ADCP group 1 breadth score, ADCP group 2 breadth score). Group 1 and group 2 points were then summed to obtain an overall ADCP activity score (ADCP group 1 + 2 breadth score).

Next, we examined the correlation between cross-reactive ADCP activity and serum levels of anti-H1 HA-specific, anti-H3 HA-specific, and anti-H1 HA stem-specific IgG. There was a significant correlation between the ADCP group 1 breadth score and anti-H1 HA head-specific total IgG and IgG1 against the vaccine strain (r = 0.4780, *p* = 0.0004; r = 0.5003, *p* = 0.0002, respectively), but no significant correlation was observed between the ADCP group 1 breadth score and anti-H1 HA head-specific total IgG and IgG1 (r = 0.0009517, *p* = 0.99480) (Figure 4A). In contrast, no correlation was observed for anti-H3 HA head-specific total IgG and IgG1 against the vaccine strain in the ADCP group 2 breadth score (r = −0.02356, *p* = 0.8710; r = −0.05861, *p* = 0.6860, respectively) (Figure 4B). There was also a significant correlation between the ADCP group 1 breadth score and anti-H1 HA stem-specific total IgG and IgG1 levels (r = 0.3844, *p* = 0.0131, and r = 0.4299, *p* = 0.0050, respectively) (Figure 4C).

These results indicated that the current influenza vaccination may induce cross-reactive IgG antibodies with Fc-mediated ADCP function against a wide range of viral strains in some individuals, although these responses tend to be weaker than those of ADCC.

### 3.5. Relationship Between Cross-Reactive Neutralization Antibody, ADCC, and ADCP Activity in Serum

Finally, to examine the relationship between various cross-reactive responses to HA-specific IgG induced by the current vaccine, we analyzed the correlation between cross-reactive neutralization, ADCC, and ADCP activities. Specifically, we first analyzed the correlations between the neutralization group 1 breadth score and the ADCC group 1 breadth score, the neutralization group 2 breadth score and the ADCC group 2 breadth score, and the combined neutralization group 1 + 2 breadth score and the ADCC group 1 + 2 breadth score (r = 0.0884, *p* = 0.5418; r = −0.0785, *p* = 0.5880; r = 0.0988, *p* = 0.4947, respectively) (Figure 5A). Similarly, no correlation was observed between the neutralization group 1 breadth score and the ADCP group 1 breadth score, the neutralization group 2 breadth score and the ADCP group 2 breadth score, or the combined neutralization group 1 + 2 breadth score and the ADCP group 1 + 2 breadth score (r = 0.0651, *p* = 0.6533; r = 0.0879, *p* = 0.5439; r = 0.1797, *p* = 0.2119, respectively) (Figure 5B). In contrast, the ADCC group 1 breadth score and ADCP group 1 breadth score, ADCC group 2 breadth score and ADCP group 2 breadth score, and combined ADCC group 1 + 2 breadth score and ADCP group 1 + 2 breadth score were all significantly correlated (r = 0.5876, *p* < 0.0001; r = 0.3089, *p* = 0.0291; r = 0.4445, *p* = 0.0012, respectively) (Figure 5C). A trend toward correlation was also observed between ADCC group 1 and ADCC group 2 breadth scores (r = 0.2064, *p* = 0.1505) (Figure 5D). Additionally, ADCP group 1 and ADCP group 2 breadth scores were significantly correlated (r = 0.2788, *p* = 0.0499) (Figure 5D).

## 4. Discussion

IAV HA proteins are the primary targets of neutralizing antibodies. However, HA genes undergo continuous evolution due to the selective pressure exerted by antibody responses, which are directed primarily toward the distal membrane receptor-binding subdomain of the molecule [4]. Antibodies induced by vaccination or infection can neutralize homologous strains within a specific subtype but have limited cross-reactivity with other IAV subtypes or antigenic variants. As a result, current influenza vaccines are effective only against a narrow range of IAV strains with HA epitopes that are antigenically similar to the vaccine strain and may be ineffective against other IAV subtypes, including potential future pandemic strains. Additionally, it is generally accepted that antibodies induced by current seasonal influenza vaccinations are not expected to provide cross-protection [31]. In fact, Grohskopf et al. reported that the vaccine effectiveness during the 2017–2018 influenza season was 62% against influenza A[H1N1] pdm09 virus, 22% against influenza A[H3N2] virus, and 50% against influenza B virus [32]. This low effectiveness has generally been attributed to an antigenic mismatch between the vaccine strain and circulating viruses. To overcome this limitation, it is necessary to develop a universal vaccine that can protect against both seasonal and potentially pandemic IAV strains [33].

In this study, we used sera from 50 individuals vaccinated against influenza strains in November 2022 to measure antibody titers against the HA head of vaccine strains H1 and H3, the primary targets of current vaccines, and the HA conformational stem, which is the primary target of several universal influenza vaccines under clinical evaluation. We also measured antibody titers against the LAH epitope, which induces non-neutralizing but cross-protective IgG antibodies that are distinct from the HA-conforming stem. The results showed that antibody titers against HA of the vaccine strains increased in all individuals after vaccination. Antibody titers against the HA stem also increased in 72% of individuals for total IgG and in 90% of individuals for IgG1 after vaccination. We subsequently examined the correlation between antibody titers against the vaccine strain HA and age or BMI, revealing no significant correlation with either parameter (Appendix A).

In a study by Yassine et al., which used an HA stem probe similar to ours to evaluate pre-vaccination sera in influenza, Ebola, and SARS DNA vaccine trials, approximately 60% of the individuals were reported to react to the HA stem [28]. In contrast, in our cohort, antibody titers against the LAH epitope showed little change before and after vaccination, which is consistent with the findings of Wang et al. [13]. This finding suggests that the current seasonal influenza vaccine alone is unlikely to induce antibodies against the LAH epitopes.

Regarding the neutralizing activity, no correlation was observed between the neutralization breath score and antibody titers against H1 or H3 head-specific IgG or H1 HA stem-specific regions. These findings are consistent with previous reports, indicating that neutralizing antibodies are effective only against IAV strains with HA epitopes that are antigenically homologous to the vaccine strain [31]. However, the neutralizing titers induced by the vaccination were divided into two distinct groups. This variability may be influenced by factors such as individual differences in the likelihood of producing neutralizing antibodies post-vaccination, the rate of antibody maturation, or age. However, further studies are needed to clarify these effects. The observation that the neutralizing titers against the previous year’s virus strains were higher than those against the vaccination strains suggests that some individuals may have been sensitized to the previous season’s influenza virus before vaccination.

In contrast, ADCC and ADCP breath scores showed high correlations in group 1, group 2, and group 1 + 2, indicating that the induction of antibodies with Fc-mediated effector functions is effective in cross-protection. The strong correlation between H1 HA stem-specific antibody titers and ADCC and ADCP group 1 breath scores reflects the characteristics of HA stem antibodies. Notably, in a study by Leon et al., HA stem-targeting antibodies are particularly effective at inducing Fc receptor-mediated effector functions such as ADCC and ADCP, suggesting a potential role in providing cross-protection against diverse influenza strains [34]. This indicates that Fc-mediated cross-reactive functions can be achieved, to some extent, even with the current influenza vaccine. However, it should be noted that the ADCC and ADCP activities in this study were evaluated in vitro. Further research is required to determine the levels of activity necessary to confer protective effects in vivo. For example, in non-human primate models (NHPs), the trivalent inactivated influenza vaccine (TIV) did not induce ADCC activity or CD8^+^ T cell responses, although binding antibodies were detected. In contrast, natural H1N1 infection has been reported to induce strong ADCC and CD8+ T cell responses against multiple H1 viral strains [35,36]. This suggests that it may be challenging to elicit cross-reactive ADCC activity in humans by vaccination with the current vaccine alone. Nevertheless, it may be possible to develop more precise criteria for ADCC and ADCP activities that can predict in vivo efficacy by monitoring the occurrence of influenza infections in individuals vaccinated with current vaccines.

Recently, efforts have been made to develop new vaccines aimed at inducing ADCC or ADCP activity through HA-specific antibody responses that cross-react with multiple groups of IAVs. Many vaccines have been specifically designed to induce antibodies against conformational epitopes of the HA stem domain. For example, a vaccine expressing a stabilized H1 HA stem antigen without an immunodominant head domain (H1ssF vaccine) has been developed using ferritin nanoparticles as a vector. Phase I clinical trials of this vaccine have been conducted in healthy adults aged 18 to 70 years [37,38]. The results showed that H1ssF vaccination induced strong plasmablast responses and sustained the induction of H1 HA stem-specific memory B cells one week post-vaccination, with a 16-fold increase in response by 2 weeks, regardless of age. Furthermore, analysis of antigen-specific antibodies induced by the vaccine revealed that they targeted two conserved epitopes on the H1 HA stem, suggesting that the vaccine-induced antibodies are capable of broadly neutralizing group 1 subtype viral strains, including strains with pandemic potential [37]. Additionally, cross-reactive neutralizing antibodies against the conserved HA stem of group 1 influenza viruses persisted and were detectable for more than a year after vaccination. ADCC activity against the H1 and H5 strains was also measured in participants’ post-vaccination sera and increased by an average of 7.3-fold at two weeks post-vaccination and remained 7.5-fold above baseline at week 18 [38].

Although this study provides valuable insights into the immune responses induced by the current seasonal influenza vaccine, it has several limitations. First, the participant sample size was relatively small, which may limit the generalizability of the findings. Second, evaluations of ADCC and ADCP activities were conducted in vitro, and the in vivo relevance of these activities remains uncertain. Third, the study focused on antibody responses, without assessing other aspects of immunity, such as T-cell responses, which also play a crucial role in protection against influenza. Fourth, this study evaluated responses against influenza A viruses, while cross-reactivity with influenza B viruses was not assessed. Finally, we did not conduct a longitudinal follow-up to evaluate the durability of the observed antibody responses. Future research should address these limitations by incorporating larger cohorts, in vivo models, and comprehensive analyses of both humoral and cellular immune responses, as well as potential cross-reactivity with influenza B viruses.

## 5. Conclusions

This study highlights the need for improved vaccines that can induce broader cross-reactivity and enhance Fc-mediated effector function. These insights contribute to the rational design of next-generation influenza vaccines with the potential to provide effective protection against both seasonal and pandemic influenza strains.

## Figures and Tables

**Figure 1 vaccines-13-00140-f001:**
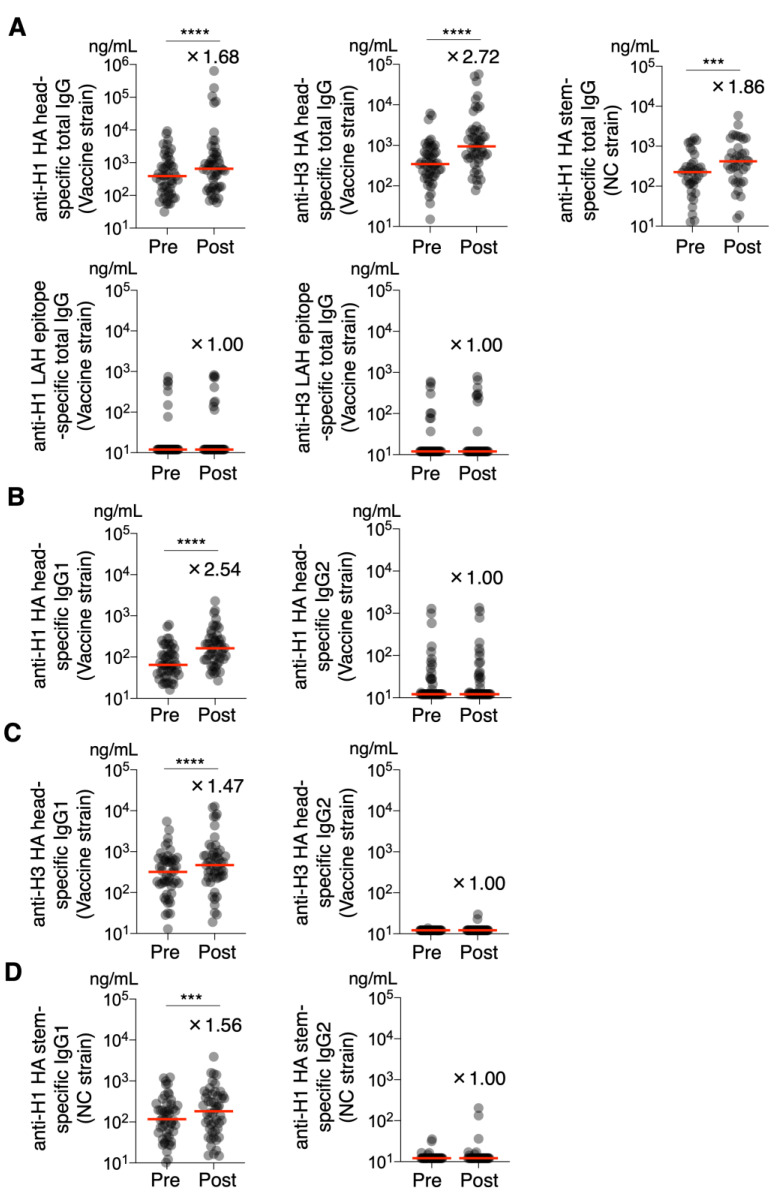
Characteristics of serum antibodies to each HA region induced by seasonal influenza vaccination. Total IgG, IgG1, and IgG2 antibodies against vaccine strains HA head-specific, H1 stem-specific, and LAH epitopes specific to vaccine strains H1 and H3 were quantified by ELISA using pre- and post-vaccination sera from seasonal influenza vaccines. Quantification was performed using ELISA. (**A**) Total IgG antibody titers against HA head-specific, H1 stem-specific, and LAH epitopes specific to vaccine strains H1 and H3 before and after vaccination. (**B**) IgG1 and IgG2 antibody titers against H1 HA head-specific before and after vaccination. (**C**) IgG1 and IgG2 antibody titers against H3 HA head-specific antibodies before and after vaccination. Total IgG, IgG1, and IgG2 antibody titers against the H1 stem before and after vaccination. (**D**) IgG1 and IgG2 antibody titers against H1 stem cells before and after vaccination. Antibody titers were quantified using standard antibodies, and the amounts are shown. Each circle represents the results for an individual participant; lines represent median values. The fold change is shown above the graph as the median antibody titer in the post-vaccination group divided by that in the pre-vaccination group. The data were analyzed using the Wilcoxon matched-pair signed-rank test. Significance is indicated by the following symbols: *** *p* < 0.001, and **** *p* < 0.0001.

**Figure 2 vaccines-13-00140-f002:**
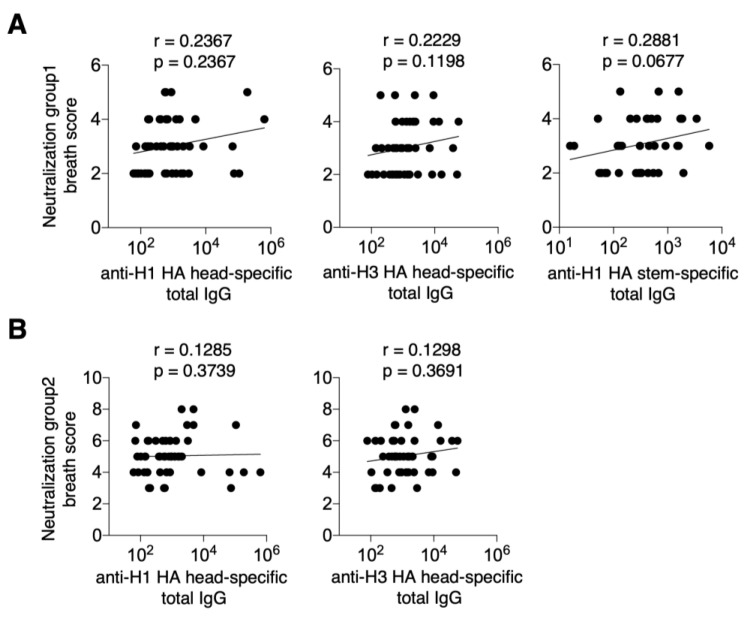
Cross-reactive neutralizing antibody activity in serum induced by current seasonal influenza vaccination. Post-vaccination sera were subjected to a microneutralization assay using ten different influenza viruses and representative data from two independent experiments. The results obtained were scored as follows: 0 for 20 and below, 1 for 21 to 80, 2 for 81 to 640, 3 for 641 to 1280, and 4 for 1280 and above. Neutralization breath scores were calculated by summing the respective scores for groups 1, 2, and group 1 + 2. Neutralization breath scores were calculated by summing the scores of groups 1, 2, and group 1 + 2. (**A**) Correlation between neutralization group 1 breath score and total IgG of H1, H3 HA head-specific and H1 stem-specific. (**B**) Correlation of neutralization group 2 breath score with total IgG for H1, H3 HA head-specific, and H1 stem-specific. Data were statistically analyzed using Spearman’s rank correlation coefficients. Lines indicate correlations determined by linear regression analysis (*n* = 50 for graphs of H1 and H3 HA head-specific antibodies and *n* = 36 for graphs of H1 stem-specific antibodies).

**Figure 3 vaccines-13-00140-f003:**
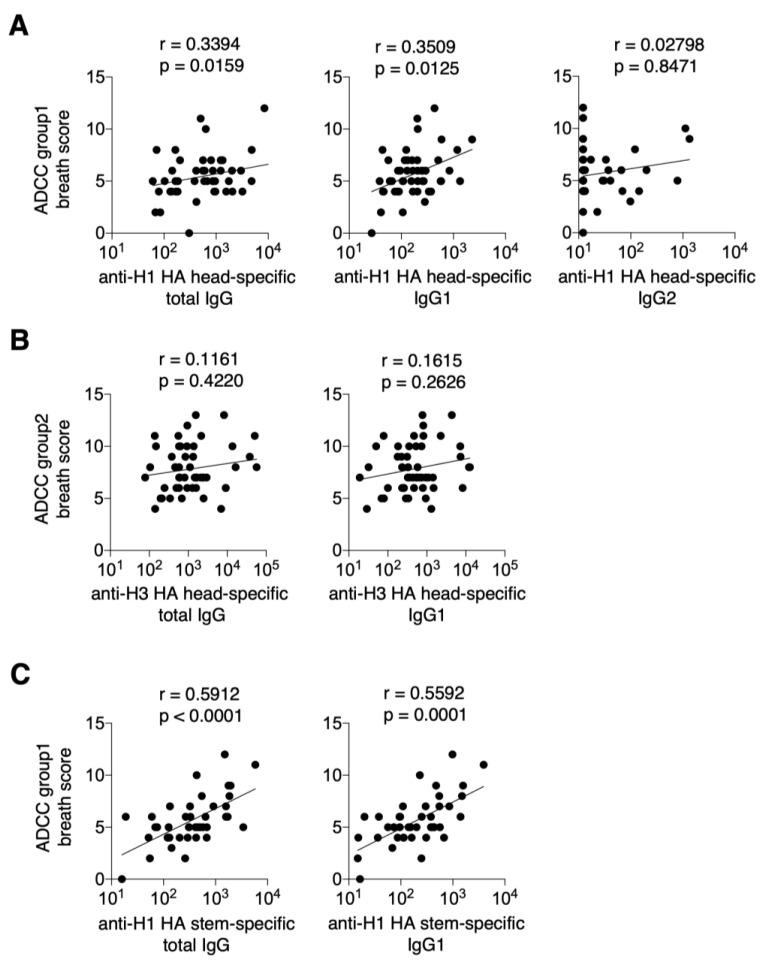
Cross-reactive ADCC activity induced by current seasonal influenza vaccination in serum. ADCC assays were performed on post-vaccination sera using 10 different influenza viruses, representative data from two independent experiments. The results obtained were scored as follows: 0 for 10 or less, 1 for 11 to 100, 2 for 101 to 500, 3 for 501 to 1000, and 4 for 1000 or more. Breath scores were calculated by summing the scores of group 1, group 2, and group 1 + 2. (**A**) Correlation between the ADCC group 1 breath score and H1 HA head-specific total IgG, IgG1, and IgG2 levels. (**B**) Correlation between the ADCC group 2 breath score and H3 HA head-specific total IgG and IgG1 levels. (**C**) Correlation between ADCC group 1 breath score and H1 stem-specific total IgG and IgG1 levels. Data were statistically analyzed using Spearman’s rank correlation coefficients. Lines indicate correlations determined by linear regression analysis (*n* = 50 for graphs of H1 and H3 HA head-specific antibodies and *n* = 36 for graphs of H1 stem-specific antibodies).

**Figure 4 vaccines-13-00140-f004:**
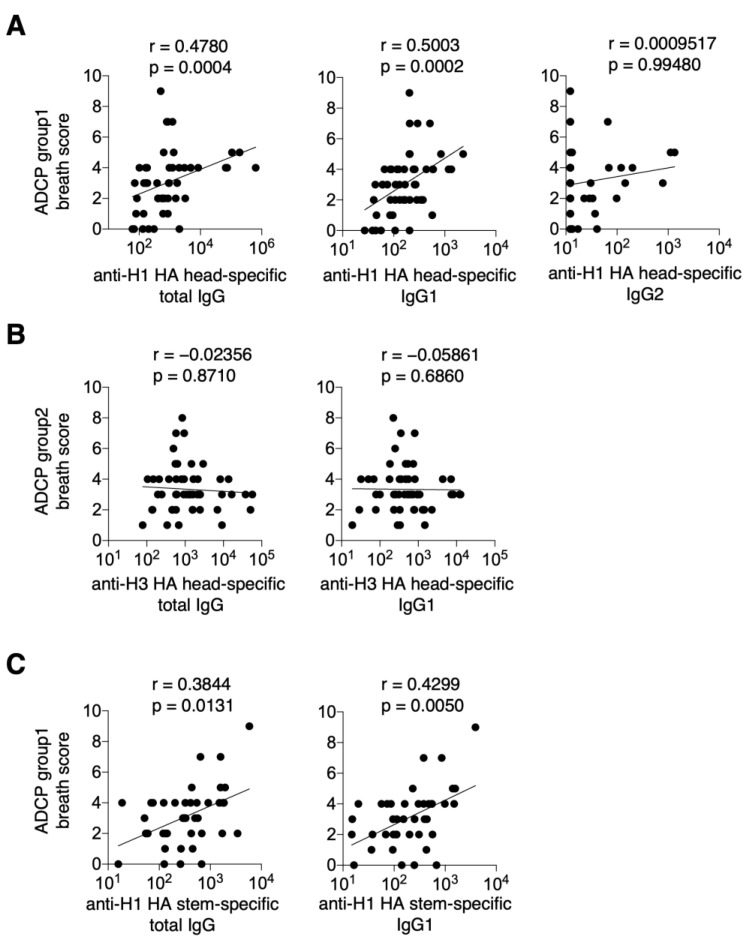
Cross-reactive ADCP activity induced by current seasonal influenza vaccination in serum. ADCP assays were performed on post-vaccination plasma with 10 different influenza viruses, and representative data were obtained from two independent experiments. The results were scored as follows: 0, 10 and below; 1, 11–100; 2, 101–500; 3, 501–1000; and 4, 1000 and above. Breath scores were calculated by summing the respective scores for groups 1, 2, and 1 + 2. (**A**) Correlation between the ADCP group 1 breath score and H1 HA head-specific total IgG, IgG1, and IgG2 levels. (**B**) Correlation between ADCP group 2 breath score and H3 HA head-specific total IgG and IgG1 levels. (**C**) Correlation between the ADCP group 1 breath score and H1 stem-specific total IgG and IgG1. Data were statistically analyzed using Spearman’s rank correlation coefficients. Lines indicate correlations determined by linear regression analysis (*n* = 50 for graphs of H1 and H3 HA head-specific antibodies and *n* = 36 for graphs of H1 stem-specific antibodies).

**Figure 5 vaccines-13-00140-f005:**
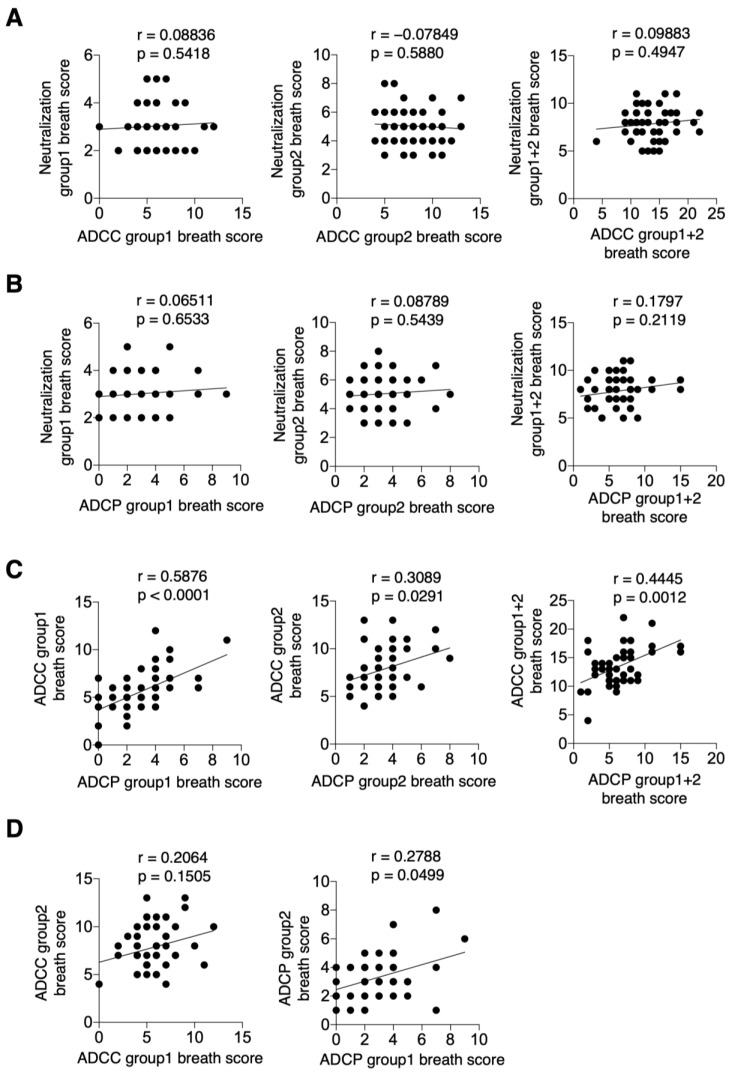
Relationship between cross-reactive neutralization antibody, ADCC and ADCP activity in serum. (**A**) Neutralization group 1 breath score and ADCC group 1 breath score, neutralization group 2 breath score and ADCC group 2 breath score, neutralization group 1 + 2 breath score and ADCC group 1 + 2 breath score, and the correlation between each breath score. (**B**) Neutralization group 1 breath score and ADCP group 1 breath score, neutralization group 2 breath score and ADCP group 2 breath score, and neutralization group 1 + 2 breath score and ADCP group 1 + 2 breath score, respectively. (**C**) ADCC group 1 breath score and ADCP group 1 breath score, ADCC group 2 breath score and ADCP group 2 breath score, and ADCC group 1 + 2 breath score and ADCP group 1 + 2 breath score, correlating with the respective breath scores. (**D**) Correlation between ADCC group 1 breath score and ADCC group 2 breath score, ADCP group 1 breath score and ADCP group 2 breath score, and their respective breath scores. Data were statistically analyzed using Spearman’s rank correlation coefficient. Lines indicate correlations determined by linear regression analysis (*n* = 50 for each graph).

## Data Availability

The data presented in this paper are available on request from the corresponding author.

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
