# Peer review of "Cross-Reactive Fc-Mediated Antibody Responses to Influenza HA Stem Region in Human Sera Following Seasonal Vaccination"

_vaccines, 2025, doi:10.3390/vaccines13020140_

Round 1
Reviewer 1 Report
Comments and Suggestions for Authors
This study examines the cross-reactivity and immunogenic characteristics of antibodies generated by standard seasonal influenza vaccines. This study concentrates on two main subjects: antibody-mediated cell death (ADCC) and phagocytosis-induced cell death (ADcp). The study design is robust, incorporating 50 participants and utilizing ELISA, neutralization, ADCC, and ADCP to assess antibody responses. The results demonstrate that HA stem-specific antibodies exhibit significant ADCC/ADCP activity; however, they are only marginally effective in eradicating vaccine-matched strains. This indicates that the current flu vaccines offer limited protection against various influenza viruses. Although seldom, self-citation indicates that authors elaborate on their prior research by referencing broader sources. Identifying in vivo evidence of ADCC/ADCP efficacy and exploring methods to enhance LAH-specific antibody responses are critical for further investigation. Nonetheless, as the study concentrated on in vitro results, it is pertinent solely for formulating frameworks for universal vaccines rather than for practical implementations.
Author Response
Point-by-point replies to the reviewers.
Firstly, we would like to the reviewers for their time and constructive and helpful suggestions. These comments have greatly helped us to strengthen the manuscript. The changes to the text made in response to the reviewer comments are highlighted in yellow.
This study examines the cross-reactivity and immunogenic characteristics of antibodies generated by standard seasonal influenza vaccines. This study concentrates on two main subjects: antibody-mediated cell death (ADCC) and phagocytosis-induced cell death (ADcp). The study design is robust, incorporating 50 participants and utilizing ELISA, neutralization, ADCC, and ADCP to assess antibody responses. The results demonstrate that HA stem-specific antibodies exhibit significant ADCC/ADCP activity; however, they are only marginally effective in eradicating vaccine-matched strains. This indicates that the current flu vaccines offer limited protection against various influenza viruses. Although seldom, self-citation indicates that authors elaborate on their prior research by referencing broader sources. Identifying in vivo evidence of ADCC/ADCP efficacy and exploring methods to enhance LAH-specific antibody responses are critical for further investigation. Nonetheless, as the study concentrated on in vitro results, it is pertinent solely for formulating frameworks for universal vaccines rather than for practical implementations.
Reply:
Thank you for your valuable comments.
We acknowledge that our study’s focus on in vitro analyses is a limitation. Further, we agree with your assessment and have emphasized this point further in the revised manuscript. While our findings are limited to in vitro results, we believe that these findings nevertheless have potential to contribute to guiding the design of universal influenza vaccines. In future studies, we plan to validate the efficacy of ADCC and ADCP mechanisms through in vivo investigations to strengthen the foundation of our conclusions. We have now added the following sentences in the final part of the discussion (Line 573, Page 16)
“Although this study provides valuable insights into the immune responses induced by the current seasonal influenza vaccine, it has several limitations. First, the participant sample size was relatively small, which may limit the generalizability of the findings. Second, evaluations of ADCC and ADCP activities were conducted in vitro, and the in vivo relevance of these activities remains uncertain. Third, the study focused on antibody responses, without assessing other aspects of immunity, such as T-cell responses, which also play a crucial role in protection against influenza. Fourth, this study evaluated responses against influenza A viruses, while cross-reactivity with influenza B viruses was not assessed. Finally, we did not conduct a longitudinal follow-up to evaluate the durability of the observed antibody responses. Future research should address these limitations by incorporating larger cohorts, in vivo models, and comprehensive analyses of both humoral and cellular immune responses, as well as potential cross-reactivity with influenza B viruses.”

Reviewer 2 Report
Comments and Suggestions for Authors
This manuscript focuses on cross-reactive Fc-mediated antibody responses targeting the conserved hemagglutinin (HA) stem region. Using various assays, including neutralization, ADCC, and ADCP, the study comprehensively evaluates immune responses following vaccination. The findings on HA stem-specific IgG1 and its correlation with Fc-mediated functions (ADCC/ADCP) provide valuable insights for the design of universal influenza vaccines.
The methods section of the manuscript indicates that the study involved only 50 participants from a single region. For studies of this type, the sample size and diversity of the study population are limited. It is recommended that the authors address this limitation in the discussion section of the manuscript.
Additionally, the methods section does not specify the procedures for serum preparation and storage. This information is crucial for experiments based on immunological methods. It is recommended that the authors provide details on these processes in the methods section.
The antibody levels in participants after vaccination are significantly influenced by factors such as age and health status. Including this type of information in the manuscript would make it more comprehensive. It is recommended that the authors consider adding these details to enhance the study's context and analysis.
Author Response
Point-by-point replies to the reviewers.
Firstly, we would like to the reviewers for their time and constructive and helpful suggestions. These comments have greatly helped us to strengthen the manuscript. The changes to the text made in response to the reviewer comments are highlighted in yellow.
This manuscript focuses on cross-reactive Fc-mediated antibody responses targeting the conserved hemagglutinin (HA) stem region. Using various assays, including neutralization, ADCC, and ADCP, the study comprehensively evaluates immune responses following vaccination. The findings on HA stem-specific IgG1 and its correlation with Fc-mediated functions (ADCC/ADCP) provide valuable insights for the design of universal influenza vaccines.
The methods section of the manuscript indicates that the study involved only 50 participants from a single region. For studies of this type, the sample size and diversity of the study population are limited. It is recommended that the authors address this limitation in the discussion section of the manuscript.
Reply:
Thank you for pointing out this important limitation. We have addressed this concern by adding the following statement to the discussion (Line 574, Page 16):
“First, the participant sample size was relatively small, which may limit the generalizability of the findings.”
Additionally, the methods section does not specify the procedures for serum preparation and storage. This information is crucial for experiments based on immunological methods. It is recommended that the authors provide details on these processes in the methods section.
Reply:
In accordance with the reviewer's comment, we have now added the following sentences in the section 2.1. (Line 115, Page 3)
“Samples were processed using the INSEPACK II-D (SEKISUI MEDICAL CO., LTD., Japan) kit to separate the serum, which was subsequently stored at −80°C.”
The antibody levels in participants after vaccination are significantly influenced by factors such as age and health status. Including this type of information in the manuscript would make it more comprehensive. It is recommended that the authors consider adding these details to enhance the study's context and analysis.
Reply:
Thank you for your valuable comments. In response to your concern, we have examined the correlation between antibody titers against the vaccine strain HA and age or BMI. However, this analysis revealed no significant correlation with either parameter, and we determined that these factors therefore do not need to be considered in the current study. The results of this analysis have now been included as a new supplemental figure (Supplemental Figure 4) in the revised manuscript. We have further added the following sentence in the discussion section (Line 512, Page 15):
“We subsequently examined the correlation between antibody titers against the vaccine strain HA and age or BMI, revealing no significant correlation with either parameter (Supplemental Figure 4).”

Reviewer 3 Report
Comments and Suggestions for Authors
The authors, in this paper, address an important aspect of influenza vaccine research by evaluating cross-reactive immune responses targeting the conserved HA stem region. While the study aims to provide insights into the potential for broad cross-protection through Fc-mediated functions such as ADCC and ADCP, the manuscript, in its current form, contains several critical shortcomings that preclude acceptance.
The materials and methods section lacks clarity and consistency. The authors do not specify whether peptides or full-length proteins were used for the LAH region, which is crucial for interpreting the findings. The use of peptides, if applied, might explain the lower IgG1 responses observed, as peptides often fail to mimic the native antigenic structure of proteins. This discrepancy is not addressed, raising concerns about the validity of the conclusions.
Furthermore, the authors fail to provide adequate information about the vaccine used, including its manufacturer, formulation, and composition. The virus strains included in the functional assays are not explicitly linked to the vaccine strains, making it difficult to assess the relevance of the data. The omission of influenza B strains, which are a standard component of seasonal influenza vaccines, further weakens the study’s comprehensiveness.
To strengthen the manuscript, the authors must clarify the antigen preparation and explicitly connect the virus strains used in the assays to the vaccine formulation. Additionally, while in vivo validation is not a requirement for the current study, including such data in future revisions could enhance the robustness and translational relevance of the findings.
In its current form, the paper requires substantial revisions to address these critical issues before it can be considered for publication.
Author Response
Point-by-point replies to the reviewers.
Firstly, we would like to the reviewers for their time and constructive and helpful suggestions. These comments have greatly helped us to strengthen the manuscript. The changes to the text made in response to the reviewer comments are highlighted in yellow.
The authors, in this paper, address an important aspect of influenza vaccine research by evaluating cross-reactive immune responses targeting the conserved HA stem region. While the study aims to provide insights into the potential for broad cross-protection through Fc-mediated functions such as ADCC and ADCP, the manuscript, in its current form, contains several critical shortcomings that preclude acceptance.
The materials and methods section lacks clarity and consistency. The authors do not specify whether peptides or full-length proteins were used for the LAH region, which is crucial for interpreting the findings. The use of peptides, if applied, might explain the lower IgG1 responses observed, as peptides often fail to mimic the native antigenic structure of proteins. This discrepancy is not addressed, raising concerns about the validity of the conclusions.
Reply:
Thank you for your detailed and constructive comments. To clarify, we used peptides rather than full-length proteins, to specifically evaluate the LAH region, as the latter would also include non-specific regions, potentially confounding the results. We have clarified this in Section 2.3 (Line 154, Page 4) of the revised manuscript.
In this study, we used LAH sequences from H1N1 (A/Guangdong-Maonan SWL1536/2019) and H3N2 (A/Hong Kong/2671/2019). To enhance clarity and provide more detailed information, we have included the LAH sequences used in the study in a new Supplementary Table (Table S2). This addition has been noted in the revised manuscript to ensure the transparency and reproducibility of our work.
Furthermore, the authors fail to provide adequate information about the vaccine used, including its manufacturer, formulation, and composition. The virus strains included in the functional assays are not explicitly linked to the vaccine strains, making it difficult to assess the relevance of the data. The omission of influenza B strains, which are a standard component of seasonal influenza vaccines, further weakens the study’s comprehensiveness.
To strengthen the manuscript, the authors must clarify the antigen preparation and explicitly connect the virus strains used in the assays to the vaccine formulation. Additionally, while in vivo validation is not a requirement for the current study, including such data in future revisions could enhance the robustness and translational relevance of the findings.
In its current form, the paper requires substantial revisions to address these critical issues before it can be considered for publication.
Reply:
Thank you for your detailed and constructive comments. We have now added the following sentences to clarify the information about the vaccines used (Line 118, Page 3).
“The vaccine used in this study was administered to participants following their explicit consent. The vaccine manufacturers included Denka Co., Ltd.; the Research Foundation for Microbial Diseases of Osaka University; KM Biologics Co., Ltd.; and DAIICHI SANKYO Co., Ltd. All vaccines were egg-based, inactivated, split influenza vaccines using standardized domestic strains (A; A/Victoria/1/2020 [IVR-217] and A/Darwin/9/2021 [SAN-010] and B; B/Phuket /3073/2013 and B/Austria/1359417/2021 [BVR-26]) approved by the national regulatory authority. There were no differences in formulation among the vaccines, and the method of administration was confirmed to be consistent across all participants.”
Furthermore, the virus strains used in the functional assays were consistent with the vaccine strains, ensuring the relevance of the data. This information has been explicitly stated in Section 2.2 (Line 140, Page 3) as follows:
“A/Victoria/1/2020 (IVR-217) and A/Darwin/9/2021 (SAN-010) are identical to the strains included in the administered vaccine.”
In this study, we focused on evaluating cross-reactivity within influenza A virus strains. While we acknowledge the importance of investigating cross-reactivity with influenza B virus strains, due to resource limitations, this was beyond the scope of the current study. We have noted this limitation in the discussion and highlighted its significance for future research (Line 579, Page 16).
“Fourth, this study evaluated responses against influenza A viruses, while cross-reactivity with influenza B viruses was not assessed. Finally, we did not conduct a longitudinal follow-up to evaluate the durability of the observed antibody responses. Future research should address these limitations by incorporating larger cohorts, in vivo models, and comprehensive analyses of both humoral and cellular immune responses, as well as potential cross-reactivity with influenza B viruses.”
Details regarding the antigens used in the ELISA have been added to Section 2.3 (Line 148, Page 4) of the revised manuscript for clarity and consistency.
“The rHA proteins used in the ELISA assays were produced in-house. Specifically, synthetic cDNA for each antigen was obtained through artificial gene synthesis and inserted into a plasmid under the control of a CMV promoter. This plasmid was subsequently transfected into Expi293 cells, which were cultured for five days post-transfection. After the culture period, the supernatants were collected, and the target antigen proteins were purified using His tag affinity chromatography.”
We hope that these revisions and clarifications adequately address the reviewer’s concerns and enhance the manuscript's clarity and comprehensiveness. We are grateful for your feedback, which has helped us improve the quality of our work.

Round 2
Reviewer 3 Report
Comments and Suggestions for Authors
The authors have addressed the key concerns regarding clarity, consistency, and comprehensiveness of the materials and methods. While the omission of influenza B strains and in vivo validation remain limitations, the authors have provided satisfactory explanations and included these aspects in the discussion as areas for future improvement.